# Clinical Implications of Metabolic Syndrome in Psoriasis Management

**DOI:** 10.3390/diagnostics14161774

**Published:** 2024-08-14

**Authors:** Maria-Lorena Mustata, Carmen-Daniela Neagoe, Mihaela Ionescu, Maria-Cristina Predoi, Ana-Maria Mitran, Simona-Laura Ianosi

**Affiliations:** 1Doctoral School, University of Medicine and Pharmacy of Craiova, 200349 Craiova, Romania; umlorena@yahoo.com (M.-L.M.); dearanamaria@yahoo.com (A.-M.M.); 2Department of Internal Medicine, Faculty of Medicine, University of Medicine and Pharmacy of Craiova, 200349 Craiova, Romania; 3Department of Medical Informatics and Biostatistics, University of Medicine and Pharmacy of Craiova, 200349 Craiova, Romania; 4Department of Morphology, Faculty of Medicine, University of Medicine and Pharmacy of Craiova, 200349 Craiova, Romania; predoi.cristina@yahoo.com; 5Department of Dermatology, Faculty of Medicine, University of Medicine and Pharmacy of Craiova, 200349 Craiova, Romania; simonaianosi@hotmail.com

**Keywords:** psoriasis, metabolic syndrome, IL-17A, insulin resistance

## Abstract

Psoriasis is an increasingly common chronic immune-mediated skin disease recognized for its systemic effects that extend beyond the skin and include various cardiovascular diseases, neurological diseases, type 2 diabetes, and metabolic syndrome. This study aimed to explore the complex relationship between psoriasis and metabolic syndrome by analyzing clinical, biochemical, and immunological parameters in patients with psoriasis alone and in patients combining psoriasis and metabolic syndrome. A total of 150 patients were enrolled, 76 with psoriasis only (PSO) and 74 with psoriasis and metabolic syndrome (PSO–MS). Data collected included anthropometric measurements, blood tests, and inflammatory markers. Statistical analysis was performed using the independent *t*-test, Mann–Whitney U test, Kruskal–Wallis test, and chi-square test to compare the two groups. Patients in the PSO–MS group had a significantly higher body weight, abdominal circumference, BMI, and inflammatory markers compared to patients with PSO. In addition, increased levels of IL-17A, cholesterol, triglycerides, and glucose were observed in the PSO–MS group. This study highlights the increased metabolic risk and exacerbated systemic inflammation associated with the coexistence of psoriasis and metabolic syndrome. These findings demonstrate the need for a comprehensive therapeutic approach and early intervention to manage metabolic complications in patients with psoriasis and metabolic syndrome.

## 1. Introduction

Psoriasis is a widespread, immune-mediated, chronic inflammatory skin condition that affects between 1% and 3% of the global population, including approximately 3% of adults and 0.1% of children and adolescents in the United States [1]. It is typically characterized as red, scaly, clearly defined plaques, particularly on the elbows, scalp, knees, and lower back, although manifestations can occur on any portion of the skin. A combination of genetic, epigenetic, environmental, and lifestyle factors determine the onset and exacerbations of the disease [2]. The prevalence of psoriasis differs significantly between different regions and ethnic groups, with the lowest reported incidence being 0.17% in East Asia and the highest being 8% in Scandinavia. Psoriasis symptoms usually appear in early adulthood, affecting both sexes approximately equally, with a frequent onset between 20 and 30 years of age [3].

Understanding the underlying mechanisms of psoriasis remains a challenge because of its complexity. Recent research has shown that psoriasis extends beyond being a simple skin condition, therefore having a systemic impact on the whole organism. Due to immune system malfunctions, this condition induces persistent low-grade inflammation, predisposing individuals to various additional health issues. Such comorbidities often include cardiovascular conditions, psoriatic arthritis, inflammatory bowel disease, depression, neurological complications, type 2 diabetes, and metabolic syndrome [4,5].

The impact of psoriasis on patients’ psychological and social well-being is considerable. The visibility of the disease often leads to stigmatization, social isolation, and psychological distress. Recent research has highlighted the profound influence of psoriasis on significant life decisions and overall quality of life. For example, a cross-sectional study conducted by Sanchez-Diaz et al. [6] delved into the factors affecting major life-changing decisions among individuals with psoriasis. The study suggested that the psychological and social burdens of the condition were substantial, which profoundly impacted patients’ choices in areas such as career, relationships, and social engagements.

Metabolic syndrome is characterized by the combination of various interconnected pathological conditions that increase the likelihood of experiencing serious cardiovascular events, including atherosclerotic coronary artery disease and cerebrovascular accidents [7,8]. It includes a combination of well-known cardiovascular risk factors, particularly central obesity, dyslipidemia, hypertension, and low glucose tolerance. There is a clear link between psoriasis and metabolic syndrome, both of which are characterized by systemic inflammation and increased levels of inflammatory biomarkers [9].

Metabolic syndrome is often associated with obesity, as adipose tissue acts as an active endocrine organ and releases substances that stimulate inflammation and attract immune cells, turning macrophages into pro-inflammatory M1 phenotypes. Some of the most important adipokines are TNF-alpha, IL-6, and leptin. Leptin specifically affects T cells, macrophages, and other immune cells, triggering a wide range of cytokine responses. This includes the production of resistin, chemerin, fibrinogen, and C-reactive protein (CRP), while simultaneously reducing levels of the anti-inflammatory adipokine, adiponectin. The biochemical activity of these substances contributes to imbalances in blood glucose levels, elevated lipid levels, and impaired blood vessel function [10,11,12].

In recent years, there has been an increasing recognition of the active endocrine role of the adipose tissue, influencing a wide range of metabolic processes. Adipocytes secrete a variety of bioactive substances called adipokines, such as cytokines, hormones, and growth factors [13]. Adipokines are pivotal in maintaining internal homeostasis, regulating blood glucose levels, managing lipid metabolism, and controlling blood pressure. Additionally, they also play a role in controlling insulin sensitivity and participate in inflammatory and immune processes. Furthermore, the role of adipokines in the development of metabolic syndrome has been deliberated [14].

In order to diagnose metabolic syndrome, at least three of the five criteria must be present: fasting glucose levels higher than 100 mg/dL, or undergoing treatment for high blood sugar; blood pressure > 130/85 mmHg, or undergoing treatment for arterial hypertension; serum triglyceride levels higher than 150 mg/dL, or undergoing treatment for high triglycerides; low levels of high-density lipoprotein (HDL) serum, with less than 40 mg/dL in males and less than 50 mg/dL in females, or undergoing treatment for dyslipidemia; and abdominal circumference of 102 cm or more in males, 88 cm or more in females [15]. 

## 2. Materials and Methods

### 2.1. Patient Selection

This retrospective cross-sectional study was conducted in the Department of Dermatology of the Emergency Clinical Hospital of Craiova and Dr Ianosi Private Medical Center between May 2022 and December 2023, and included all patients diagnosed with psoriasis alone, and both psoriasis and metabolic syndrome. The criteria for inclusion in the study were confirmed diagnosis of moderate or severe psoriasis according to the American Academy of Dermatology criteria and the presence of metabolic syndrome diagnosed based on NHLBI and AHA criteria, requiring the presence of at least three of the following five conditions: fasting blood glucose > 100 mg/dL or treatment for hyperglycemia, blood pressure > 130/85 mmHg or treatment for hypertension, serum triglycerides > 150 mg/dL or treatment for hypertriglyceridemia, low HDL levels (men < 40 mg/dL, women < 50 mg/dL) or treatment for dyslipidemia, and abdominal circumference ≥ 102 cm in men or ≥88 cm in women. Exclusion criteria from the study were the presence of other autoimmune diseases or severe systemic diseases other than metabolic syndrome to ensure a focused analysis of the relationship between psoriasis and metabolic syndrome.

The selection process strictly adhered to ethical guidelines. Informed consent was obtained from all participants, and the study was conducted in compliance with the Declaration of Helsinki (2004). This study was conducted in accordance with the Declaration of Helsinki and approved by the Institutional Review Board of University of Medicine and Pharmacy Craiova (no. 195/20.09.2022).The ethical considerations ensured that participants were fully informed about the study’s objectives, procedures, and potential risks, thereby upholding the integrity of the research and the safety of the involved patients.

We have gathered personal information, medical history, and various health data for all patients, including anthropometric measurements such as height, weight, body mass index (BMI), and abdominal circumference (AC), as well as biological parameters including total blood count, glycemia, total cholesterol, LDL and HDL cholesterol fractions, triglycerides, transaminases (ALT, AST), leptin, IL-17, IL-23, and C-reactive protein (CRP).

### 2.2. Statistical Analysis

Statistical tests were applied using the SPSS (Statistical Package for Social Sciences) software, version 20 (SPSS Inc., Chicago, IL, USA). Continuous variables were defined as the mean ± standard deviation (SD), while frequency distributions and percentages were used to describe nominal and ordinal variables. Normality distribution was assessed using the Kolmogorov–Smirnov/Shapiro–Wilk test. Correlations, and comparisons between the study group and the control group were performed with the independent *t*-test for normally distributed data, Kendall’s tau-b, Mann–Whitney U and Kruskal–Wallis tests for continuous non-normally distributed data, and chi-square test for categorical data. All *p*-values less than 0.05 were considered statistically significant.

## 3. Results

Following the application of the inclusion and exclusion criteria, the study group comprised 150 patients, 58 females (representing 38.67% from the entire study lot) and 92 males (61.33%), with ages between 37 and 76 years old (mean ± SD was 55.92 ± 9.80). All patients were previously diagnosed with psoriasis, and almost half of them (74 patients, 49.33%) also had metabolic syndrome (MS); therefore, the study group was divided in two subgroups, namely PSO (patients with psoriasis) and PSO–MS (patients with both psoriasis and MS).

### 3.1. PSO–MS and PSO Group Analysis

A Mann–Whitney U test was run to determine if there were differences in the ages of patients from both subgroups. Distributions of the ages for PSO and PSO–MS patients were similar, as assessed by visual inspection. The median age for PSO patients (53) was smaller than the median age for PSO–MS patients (55), but the difference was not statistically significantly different, U = 2628.0, z = −0.692, *p* = 0.489.

Gender distribution was statistically significantly different between the two subgroups, as females represented 34.5% from the PSO–MS subgroup, and 65.5% from the PSO subgroup (χ^2^(1) = 8.344, *p* = 0.004). AHT presence was also statistically significantly different between the two subgroups (PSO–MS and PSO), as all patients from the PSO–MS group had AHT (100%), compared to only 5.1% patients from the PSO group (χ^2^(1) = 134.818, *p* < 0.001).

Patients included in the PSO subgroup had a median weight of 79.50 kg, which was significantly lower than the weight of the patients included in the PSO–MS subgroup, who had a median weight of 102.00 kg (*p* < 0.001). Implicitly, the abdominal circumference (AC) was higher for patients with MS, with a median value of 102.0 cm, compared to PSO patients, with a median AC of 87.0 cm (*p* < 0.001). Similar results were obtained following the BMI analysis: patients included in the PSO subgroup had a median BMI of 27.41, which was significantly lower than the BMI of the patients included in the PSO–MS subgroup, who had a median BMI of 32.35 (*p* < 0.001). Patient characteristics are presented in Table 1.

For cholesterol, triglycerides, ALT, and glycaemia, patients with PSO–MS had higher values than patients with only PSO, and the differences were statistically significant (*p* < 0.05, Table 1). For AST, it was also found that patients with PSO–MS had higher values compared to patients included in the other subgroup; however, the differences were relatively small and not significant from a statistical point of view (*p* = 0.845). CRP, neutrophils, and NLR also had significantly higher values for patients included in the PSO–MS subgroup, compared to patients from the PSO subgroup (*p* < 0.05).

In contrast, HDL and lymphocytes were lower for patients with MS, compared to patients without MS, with statistically significant differences between values (*p* < 0.05). Therefore, LHR presented the same trend as the previous variables, with increased values for PSO–MS patients compared to PSO patients, and differences that were statistically significant (*p* < 0.001).

Leptin was also higher for patients within the PSO–MS subgroup, compared to PSO patients, with statistically significant differences (*p* < 0.001). Group comparisons regarding IL-17A and IL-23 yielded different results: the values for PSO–MS group were higher for both interleukins, but IL-17A presented statistically significantly differences compared to the PSO group (*p* = 0.009), while IL-23 has similar values for both groups.

PASI, the main indicator of psoriasis, has statistically significantly higher values for patients within the PSO–MS group (median 20.40) compared to patients within the PSO group (median 14.90), *p* = 0.034.

### 3.2. PSO Treatment

According to the patients’ context, four different PSO treatments were administered: anti-TNF-α, anti-IL-23, anti-IL-17 and methotrexate. The distribution of patients according to the therapy type and subgroup inclusion is indicated in Figure 1, which presents a similar number of patients in both subgroups for each type of medication. A chi-square test of independence was conducted between therapy type and subgroup. All expected cell frequencies were greater than five. There was no statistically significant association between therapy and subgroup, χ^2^(2) = 3.612, *p* = 0.110. The numerical value of the association was small (Cohen, 1988), Cramer’s V = 0.201.

The treatment for PSO was started after a certain time interval from the initiation of psoriasis. The median time interval between PSO debut and treatment start was higher for PSO–MS patients compared to PSO patients (13.00 months vs. 10.5 months), but the differences between groups were not statistically significant (*p* = 0.324).

Gender distribution was not statistically significantly different between the various therapy types, (χ^2^(3) = 2.804, *p* = 0.423)—Figure 2.

Distributions of the ages for the four categories of therapy were similar, as assessed by visual inspection. The median age had an increased trend for patients treated with methotrexate (50.50 years old), to patients treated with anti-IL17 (54.00 years old), anti-IL23 (55.00 years old), or anti-TNF-α (56 years old), but the overall differences between the ages were not statistically significant, χ^2^(3) = 2.921, *p* = 0.404.

The majority of parameters acquired within our study had similar median values for patients receiving various therapies, without statistically significant differences between groups (*p* > 0.05)—Table 1.

Distributions of leptin values were similar for all groups, as assessed by visual inspection of a boxplot. Median leptin values were statistically significantly different between the different therapy groups, χ^2^(3) = 10.212, *p* = 0.017. Subsequently, pairwise comparisons were performed using Dunn’s (1964) procedure. A Bonferroni correction for multiple comparisons was made with statistical significance accepted at the *p* < 0.0083 level. This post-hoc analysis revealed statistically significant differences in leptin values between the anti-IL-17 (519.61) and methotrexate (1134.13) groups (*p* = 0.017), the anti-IL-23 (681.50) and methotrexate groups (*p* = 0.036), and between the anti-TNF-α (725.01) and methotrexate groups (*p* = 0.048), but not between any other group combination. 

The distribution of IL-17 had a similar behavior, with median values being statistically significantly different between the different therapy groups, χ^2^(3) = 11.316, *p* = 0.010. The post-hoc analysis revealed statistically significant differences between the same groups as for leptin.

Other statistically significant differences between the therapy types were identified for glycaemia, NLR, and lymphocytes (*p* < 0.05 for all three variables—Table 2). For glycaemia, the methotrexate group had the highest median value (144.50), while for NLR, the highest median value (2.81) was for the IL-23 group, and for lymphocytes, the highest median value (28.85) was recorded for the IL-17 group.

PASI median values increased from 14.40 for the anti-IL-23 group, to 15.60 for anti-IL-17, 22.30 for methotrexate, and 22.60 anti-TNF-α; however, they did not differ significantly between the four types of therapy, even if the *p*-value was close to the threshold (*p* = 0.057).

### 3.3. PASI Values

For all patients, the PASI values were measured before the initiation of PSO treatment. A Mann–Whitney U test was run to determine if there were differences in PASI values between males and females. Distributions of PASI values for males and females were similar, as assessed by visual inspection. The median PASI for males (18.20) was higher than for females (14.40), yet the difference was not statistically significantly different, U = 3036.00, z = 1.421, *p* = 0.155. A similar test was run to determine whether there were differences in PASI values between patients with and without AHT. Distributions of PASI values for those two categories of patients were similar, as assessed by visual inspection. The median PASI value was not statistically significantly higher in patients with AHT (20.40) than in patients without AHT (14.90), U = 2330.00, z = 01.799, *p* = 0.072.

A Kendall’s tau-b correlation was run to determine the relationship between PASI values and the time interval between PSO debut and the initiation of therapy (expressed in months). There was a moderate, positive association between PASI values and the time interval, which was statistically significant, τb = 0.257, *p* < 0.001 (the higher the time interval between PSO diagnosis and therapy initiation, the higher the PASI value). Kendall’s tau-b correlations were also run to determine the relationship between PASI level, and the clinical parameters included in Table 1, amongst 150 participants. There were mostly weak-to-moderate positive associations between the PASI index at the beginning of PSO treatment and the parameters included in Table 1, and most tests were statistically significant (Table 3).

Weight (kg) seemed to be positively correlated to PASI values too, in a significant way, τb = 0.194, *p* < 0.001, in addition to cholesterol, TG, and glycaemia, while smaller values of HDL were correlated to higher values of the PASI (τb = −0.135, *p* = 0.016). All other parameters were not correlated to PASI values (*p* > 0.05).

## 4. Discussion

Several studies have indicated that metabolic syndrome occurs more frequently in patients with psoriasis [16,17,18]. The correlation between these two conditions is primarily attributed to shared inflammatory pathways that involve cytokines such as TNF-a, IL-6, and IL-17, which significantly contribute to the pathogenesis of both diseases [18,19,20,21]. These cytokines disrupt insulin signaling pathways by promoting serine phosphorylation of insulin receptor substrate-1 (IRS-1); increase lipid metabolism disturbances, leading to the release of free fatty acids from adipose tissue; and impair endothelial function [22,23].

Moreover, additional research has illustrated a correlation between an increased body mass index (BMI), hip and waist circumference, and insulin levels, the key components in metabolic syndrome, with the severity of psoriasis disease [24]. 

It is crucial to understand the shared pathophysiological elements of psoriasis and metabolic syndrome in order to comprehend their association. Throughout this study, we found a strong and complex link between psoriasis (PSO) and metabolic syndrome (MS), examining clinical, biochemical, and immunological parameters in patients with psoriasis only (PSO) and those with psoriasis and metabolic syndrome at the same time (PSO–MS), which is similar to most previous studies [24,25].

The intricate mechanism linking these two conditions is currently under investigation. They both involve a complex interplay of various inflammatory and cytokine-driven processes. Recent discoveries indicate that the overproduction of proinflammatory mediators in psoriasis can permeate into the systemic circulation, leading to systemic insulin resistance, endothelial dysfunction, increased angiogenesis, heightened oxidative stress, and hypercoagulation, all of which contribute to cardiovascular damage [26]. Our research, which echoes the existing literature, highlights the significant connection between psoriasis and metabolic syndrome. This connection, along with the previously identified associations with obesity, dyslipidemia, and hypertension in patients with psoriasis, could have profound implications for the management and treatment of these conditions [27].

First of all, we found significant differences in gender distribution between the PSO and PSO–MS groups that is particularly interesting. Females were more prevalent in the PSO subgroup (65.5%), while males predominated in the PSO–MS subgroup. This finding aligns with previous studies, indicating that males with psoriasis have a higher vulnerability to metabolic syndrome [28,29]. Hormonal disparities, genetic predispositions, and lifestyle factors could contribute to this discrepancy across genders. Although the median age difference between the groups was not statistically significant, the slightly higher median age in the PSO–MS group suggests that metabolic comorbidities might develop or become more pronounced as psoriatic patients age, which highlights the need for vigilant metabolic monitoring and early intervention in older psoriatic patients to prevent the onset of metabolic syndrome. 

The findings of this study align with previous research, indicating a strong link between metabolic syndrome and psoriasis vulgaris. Numerous studies have consistently demonstrated a higher prevalence of metabolic syndrome components, such as obesity, dyslipidemia, hypertension, and insulin resistance, in individuals with psoriasis vulgaris compared to the general population [30,31,32,33,34]. Our study’s data further validate these associations, supporting the theory of a complex pathophysiological connection between the two conditions. Specifically, this research corroborates prior discussions in the literature, confirming that lipid profiles, weight, and glucose levels are notably elevated in cases where metabolic syndrome coexists with psoriasis. This suggests that metabolic syndrome exacerbates patients’ overall health profile in this context [35,36,37,38,39,40,41].

Inflammatory markers such as CRP, neutrophils, and the neutrophil-to-lymphocyte ratio (NLR) were significantly elevated in the patients from the PSO–MS subgroup, compared to patients from the PSO subgroup (*p* < 0.001). These findings indicate a heightened inflammatory state in these patients, which could contribute to the pathogenesis of both psoriasis and metabolic syndrome. The increased CRP levels, a ubiquitous inflammatory marker, align with the understanding that metabolic syndrome constitutes a pro-inflammatory state [12,42,43].

CRP, an acute-phase protein produced by the liver in response to inflammation, serves as a sensitive marker of systemic inflammation and has been extensively researched in the context of psoriasis [44,45,46]. In our study, we found that median CRP levels were markedly higher in the PSO–MS group (19.00) than in the PSO group (5.00) (*p* < 0.001).

Prior studies have consistently shown elevated CRP levels in patients with psoriasis compared to healthy controls. For example, a study by Coimbra et al. [47] reported significantly higher CRP levels in psoriatic patients, suggesting that CRP is a reliable marker for assessing systemic inflammation in these patients. Similarly, CRP levels are known to be even higher in psoriatic patients with concurrent MS. Holmannova et al. [48] noted significantly elevated CRP levels in psoriasis patients with MS compared to those without MS, aligning with our findings.

The neutrophil-to-lymphocyte ratio (NLR) is increasingly recognized as a valuable indicator of systemic inflammation and immune response [49]. It holds significant implications for various chronic inflammatory conditions, such as psoriasis and metabolic syndrome, both of which are characterized by persistent inflammation and have been associated with elevated NLR values. The elevated NLR observed in PSO–MS patients can be attributed to various contributing factors. Primarily, metabolic syndrome is characterized by chronic low-grade inflammation, which is manifested in elevated neutrophil counts [50]. Neutrophils play a crucial role as the first line of defense in response to inflammation and infection, and their increased level signifies a constant inflammatory response. This intensified state of inflammation in PSO–MS patients exacerbates psoriasis, leading to more-severe clinical manifestations, as evidenced by the higher Psoriasis Area Severity Index (PASI) scores in this particular patient group. Moreover, the elevated NLR in patients with psoriasis and metabolic syndrome may be influenced by concurrent comorbidities like obesity, hypertension, and dyslipidemia, all of which contribute to systemic inflammation [51]. A recent study conducted by Hong et al. [52] explored the link between NLR and psoriasis, finding a notable association between an elevated NLR and psoriasis and suggesting the potential of NLR as an inflammatory biomarker. 

Patients with psoriasis and metabolic syndrome are at a greater risk of developing non-alcoholic fatty liver disease (NAFLD) [46,53,54,55]. The multidimensional interactions between psoriasis, metabolic syndrome, and NAFLD involve multiple aspects. Chronic inflammation, a characteristic of both psoriasis and metabolic syndrome, promotes hepatic steatosis and progression to non-alcoholic steatohepatitis (NASH) [51,52]. Insulin resistance, a hallmark of metabolic syndrome, plays a significant role to hepatic fat accumulation. Moreover, dyslipidemia, commonly associated with metabolic syndrome, supplementarily exacerbates liver fat accumulation. Several studies have demonstrated that psoriasis patients have a higher prevalence and severity of NAFLD compared to the general population. Neagoe et al. [56] revealed that patients presenting with both psoriasis and NAFLD exhibited elevated transaminase (ALT, AST) levels compared to those with NAFLD alone. This observation may be attributed to the exacerbation of inflammation due to the presence of psoriasis, resulting in a supplementary increase in transaminase levels. Furthermore, both psoriasis and NAFLD are linked to insulin resistance, a key feature of metabolic syndrome [51]. Insulin resistance augments the influx of free fatty acids to the liver, thereby promoting hepatic fat accumulation and inflammation. 

Leptin is an adipokine primarily synthesized by adipose tissue and plays a crucial role in regulating energy balance, also exerting important immunomodulatory functions [54]. Leptin influences T cells, macrophages, and various other immune cells, eliciting a broad spectrum of cytokine responses, including resistin, chemerin, fibrinogen, and CRP, while reducing levels of adiponectin, an anti-inflammatory adipokine [55]. Research indicates that leptin may have a direct impact on endothelial cells, influencing vascular function and contributing to the development of atherosclerosis [56]. This is particularly significant in light of the heightened cardiovascular risk associated with both psoriasis and metabolic syndrome. Our study revealed that individuals in the PSO–MS group exhibited elevated leptin levels, which correlated with increased inflammatory markers such as CRP and the neutrophil-to-lymphocyte ratio (NLR). These findings suggest that leptin could serve as a significant mediator in the interaction between systemic and local inflammation in the context of psoriasis and metabolic syndrome. Therefore, the elevated levels of leptin observed in the PSO–MS group are consistent with the existing literature and may indicate a more prominent inflammatory response and a pathological connection between the systemic inflammation associated with metabolic syndrome and the localized inflammation characteristic of psoriasis [55,57,58]. 

Interleukins 17 (IL-17) and 23 (IL-23) are pro-inflammatory cytokines that play an essential role in the pathogenesis of various autoimmune diseases, including psoriasis, by promoting keratinocyte activation and proliferation [59,60,61,62,63]. These two cytokines function synergistically as a crucial axis in the immune response, rather than acting independently. One interesting and frequently underestimated aspect of IL-17 is its ability to not only induce localized inflammation in the skin, but also to contribute to systemic inflammation [64,65,66]. In our research, IL-17 levels were significantly higher in patients with both psoriasis and metabolic syndrome compared to those with only psoriasis, which suggest a more pronounced inflammatory response, potentially establishing a connection between the systemic inflammation associated with metabolic syndrome and the localized inflammation characteristic of psoriasis [59]. This association implies that IL-17 may serve as a crucial mediator in the intersection of these conditions. The significant elevation of IL-17 indicates an escalated inflammatory state in PSO–MS patients, potentially contributing to the exacerbation of both psoriasis and metabolic syndrome.

Contrary to IL-17, IL-23 levels did not show a significant difference between the PSO–MS and PSO groups. Despite IL-23 being crucial for the maintenance and expansion of Th17 cells, the similar levels in both groups suggest that its role might not differ significantly between patients with only psoriasis and those with both conditions. A recent meta-analysis carried out by Liu et al. [62] found no significant correlation between IL-23 levels and psoriasis development, suggesting that circulating inflammatory cytokine levels do not necessarily reflect their pathogenic role.

However, our study brings a unique perspective by highlighting a significant increase in IL-17A in patients with PSO–MS, providing new insights into cytokine profiles in this comorbid condition. In contrast to other studies that observed significant differences in IL-23 levels, our results indicate a possibly more decisive role of IL-17A in the dynamics between psoriasis and metabolic syndrome [61,63].

The severity of psoriasis, as measured by the Psoriasis Area and Severity Index (PASI), was higher in patients with PSO–MS compared to those with PSO only. Several studies support the correlation between psoriasis severity and metabolic syndrome. For instance, a study conducted in Jakarta [64] found that patients with severe psoriasis (PASI > 10) were significantly more likely to have metabolic syndrome, with a 3.67 times higher likelihood compared to those with milder forms of psoriasis, highlighting the increased metabolic burden in these patients. In addition, a comprehensive meta-analysis examining data from 35 studies involving over 1.45 million participants and including 46,714 psoriasis patients corroborated these findings, revealing that having severe forms of psoriasis increased the likelihood of having metabolic syndrome as well [65]. Specifically, the adjusted odds ratio for metabolic syndrome in severe psoriasis was 1.98, in contrast to 1.22 for mild psoriasis, demonstrating a dose–response relationship between the severity of psoriasis and metabolic dysfunction. This analysis supports the notion that severe psoriasis exacerbates metabolic dysfunction, probably due to heightened systemic inflammation.

The original aim of this study was to examine in detail the relationship between psoriasis and metabolic syndrome, focusing on the remarkable clinical, biochemical, and immunological aspects. This study compared patients with psoriasis alone and those with concurrent metabolic syndrome and showed that there is a link between the coexistence of these conditions and an elevated risk of metabolic syndrome. These findings are crucial for developing faster therapy techniques and better patient outcomes with comprehensive interventions and management approaches. Ultimately, this investigation aids in understanding the interconnection between chronic inflammatory diseases and metabolic disorders.

## 5. Conclusions

The presence of potential metabolic comorbidities underscores the necessity for an integrated approach to the management of psoriasis. It is imperative to conduct regular screening for metabolic syndrome in psoriasis patients to enable early intervention and prevent cardiovascular and metabolic complications. The strong associations between inflammatory markers and both conditions suggest that anti-inflammatory treatments targeting specific cytokines, such as IL-17A, may have potential benefits. Our study emphasizes the significant interplay between psoriasis and metabolic syndrome, highlighting the complex, multi-faceted relationship between these conditions. Our findings demonstrate that patients with psoriasis and concomitant metabolic syndrome show more pronounced clinical, biochemical, and immunological deviations compared to those with psoriasis alone.

## Figures and Tables

**Figure 1 diagnostics-14-01774-f001:**
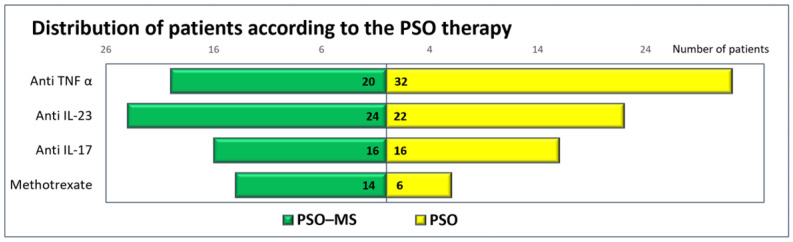
Distribution of patients according to the therapy type and study group.

**Figure 2 diagnostics-14-01774-f002:**
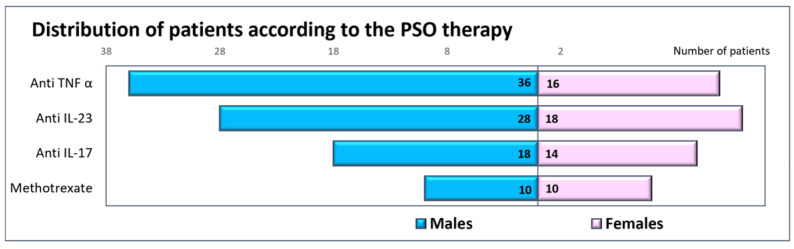
Distribution of patients according to the therapy type and gender.

**Table 1 diagnostics-14-01774-t001:** Main characteristics of the study subgroups.

Parameter	PSO–MS	PSO	*p*-Value ^1^
	Median	Median	
Weight (kg)	102.00	79.50	<0.001
AC (cm)	102.00	87.00	<0.001
BMI (kg/m^2^)	32.35	27.41	<0.001
Cholesterol (mg/dL)	230.63	219.00	0.026
Triglycerides (mg/dL)	238.00	123.55	<0.001
AST (U/L)	21.00	19.20	0.845
ALT (U/L)	26.90	21.10	0.007
Glycaemia (mg/dL)	137.93	91.28	<0.001
Leptin (ng/mL)	1131.09	481.18	<0.001
IL_17A (pg/mL)	2108.29	162.43	0.009
IL_23 (pg/mL)	195.84	221.86	0.538
C-reactive protein (mg/dL)	19.00	5.00	<0.001
Neutrophils (%)	75.80	60.00	<0.001
NLR	3.21	2.29	<0.001
Lymphocytes (%)	24.00	27.00	<0.001
HDL (mg/dl)	36.00	55.62	<0.001
LHR (lymphocytes/HDL)	0.67	0.47	<0.001
PASI	20.40	14.90	0.034

^1^ Mann–Whitney U test.

**Table 2 diagnostics-14-01774-t002:** Main characteristics of the study group, according to the therapy type.

Parameter	Anti-IL-17	Anti-IL-23	Anti-TNF-α	Methotrexate	*p*-Value ^1^
	Median	Median	Median	Median	
Weight (kg)	88.0	93.0	81.0	100.5	0.166
CA (cm)	102.0	96.0	93.5	101.0	0.227
BMI (kg/m^2^)	31.36	30.99	29.57	30.13	0.368
Cholesterol (mg/dL)	215.50	230.00	228.00	232.51	0.634
Triglycerides (mg/dL)	177.75	179.00	190.30	209.67	0.617
AST (U/L)	19.00	18.40	22.80	22.10	0.262
ALT (U/L)	21.60	23.30	24.00	24.50	0.412
Glycaemia (mg/dL)	93.85	103.10	101.30	144.50	0.011
Leptin (ng/mL)	519.61	681.50	725.01	1134.13	0.017
IL_17A (pg/mL)	197.74	160.49	185.77	2153.25	0.010
IL_23 (pg/mL)	293.65	182.13	166.38	193.60	0.250
CRP (mg/dL)	5.95	10.00	6.95	13.90	0.127
Neutrophils (%)	65.15	69.75	64.60	75.90	0.091
NLR	2.27	2.81	2.61	3.00	0.001
Lymphocytes (%)	28.85	24.00	26.00	25.00	0.001
HDL (mg/dL)	50.00	48.00	50.98	36.50	0.079
LHR	0.56	0.54	0.51	0.68	0.096
PASI	15.60	14.40	22.60	22.30	0.057

^1^ Kruskal–Wallis H test.

**Table 3 diagnostics-14-01774-t003:** Kendall’s tau-b correlation between PASI and all clinical parameters.

Parameter	Tau-b Coefficient τb	*p*-Value ^1^
Age (years old)	0.034	0.545
Weight (kg)	0.196	0.001
CA (cm)	0.074	0.244
BMI	0.186	0.001
Cholesterol	0.226	<0.001
Triglycerides	0.221	<0.001
AST	0.120	0.061
ALT	0.043	0.451
Glycaemia	0.132	0.009
Leptin	0.191	0.002
IL_17A	−0.081	0.630
IL_23	0.178	0.010
CRP	0.196	<0.001
Neutrophils	0.199	<0.001
NLR	0.205	0.010
Lymphocytes	−0.113	0.662
HDL	−0.134	0.016
LHR (lymphocytes/HDL)	0.047	0.016

^1^ Kendall’s tau-b test.

## Data Availability

The data used to support the findings of this study are available from the corresponding author upon reasonable request.

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
