# Peer review of "Clinical Implications of Metabolic Syndrome in Psoriasis Management"

_diagnostics, 2024, doi:10.3390/diagnostics14161774_

Round 1
Reviewer 1 Report
Comments and Suggestions for Authors
Results - Table 1., Table 3.
The results require amendment concerning the p-value. The correct notation should be revised from p < 0.0005 to p < 0.001, in accordance with the guidelines for reporting results following statistical analysis in biomedicine
(EQUATOR Network | Enhancing the QUAlity and Transparency Of Health Research. Basic Statistical Reporting for Articles Published in Biomedical Journals: The “Statistical Analyses and Methods in the Published Literature” or The SAMPL Guidelines” Thomas A. Lang and Douglas G. Altman
Author Response
Results - Table 1., Table 3.
The results require amendment concerning the p-value. The correct notation should be revised from p < 0.0005 to p < 0.001, in accordance with the guidelines for reporting results following statistical analysis in biomedicine.
We agree with this comment. We have, therefore, modified to emphasize this point.
Reviewer 2 Report
Comments and Suggestions for Authors
Dear Authors,
I read with interest your manuscript. It is an interesting study. I would suggest some points to improve:
1) You could make reference in the introduction to the psychosocial burden of the disease with some recent studies (e.g. Sanchez-Diaz M, Díaz-Calvillo P, Soto-Moreno A, Molina-Leyva A, Arias-Santiago S. Factors Influencing Major Life-Changing Decisions in Patients with Psoriasis: A Cross-sectional Study. Acta Derm Venereol. 2023 Oct 10;103:adv11640. doi: 10.2340/actadv.v103.11640. PMID: 37815093; PMCID: PMC10583836).
2) Line 79, a reference for the metabolic syndrome criteria is missing.
3) Line 89, the number of patients included must be written in the results section, not in the methods section (it is not part of the "methodology").
4) Methods section should be divided into: "design of the study"; "inclusion criteria"; "variables"; "statistics"; and "ethics", to make the methods clearly understandable. It should be clearly stated which type of study is (cross-sectional?).
5) Discussion is very long and redundant. It should be focused on more novel data, and probably should avoid discussing on "logical" issues. For example, those patients with Metabolic Syndrome have higher BMI and worse lipid profile (this is logical, because they are components of the definition of the MS). These parts of the discussion should be removed, and it should be centered in the interesting points (NLR, CRP, cytokines,...).
Author Response
1) You could make reference in the introduction to the psychosocial burden of the disease with some recent studies (e.g. Sanchez-Diaz M, Díaz-Calvillo P, Soto-Moreno A, Molina-Leyva A, Arias-Santiago S. Factors Influencing Major Life-Changing Decisions in Patients with Psoriasis: A Cross-sectional Study. Acta Derm Venereol. 2023 Oct 10;103:adv11640. doi: 10.2340/actadv.v103.11640. PMID: 37815093; PMCID: PMC10583836).
We appreciate your suggestion, therefore reference was added in the introduction (line 59).
2) Line 79, a reference for the metabolic syndrome criteria is missing.
We introduced a reference as mentioned (line 94).
3) Line 89, the number of patients included must be written in the results section, not in the methods section (it is not part of the "methodology").
We agree that the number of patients included should me written in the results section, therefore we have modified accordingly (line 99, 137).
4) Methods section should be divided into: "design of the study"; "inclusion criteria"; "variables"; "statistics"; and "ethics", to make the methods clearly understandable. It should be clearly stated which type of study is (cross-sectional?).
Thank you for this observation, we performed a retrospective cross-sectional study indeed, the changes have been made as suggested. (line 99)
5) Discussion is very long and redundant. It should be focused on more novel data, and probably should avoid discussing on "logical" issues. For example, those patients with Metabolic Syndrome have higher BMI and worse lipid profile (this is logical, because they are components of the definition of the MS). These parts of the discussion should be removed, and it should be centered in the interesting points (NLR, CRP, cytokines,...).
We appreciate this suggestion. We have shortened the discussion section, trying to focus on the interesting points.